# Effect of Continuous Positive Airway Pressure on Lipid Profiles in Obstructive Sleep Apnea: A Meta-Analysis

**DOI:** 10.3390/jcm11030596

**Published:** 2022-01-25

**Authors:** Baixin Chen, Miaolan Guo, Yüksel Peker, Neus Salord, Luciano F. Drager, Geraldo Lorenzi-Filho, Xiangdong Tang, Yun Li

**Affiliations:** 1Department of Sleep Medicine, Shantou University Mental Health Center, Shantou University Medical College, Shantou 515065, China; 15bxchen@stu.edu.cn; 2Sleep Medicine Center, Shantou University Medical College, Shantou 515041, China; 3Department of Nursing, Shantou University Medical College, Shantou 515041, China; mlguo@stu.edu.cn; 4Department of Pulmonary Medicine, School of Medicine, Koc University, 34010 Istanbul, Turkey; yuksel.peker@lungall.gu.se; 5Sahlgrenska Academy, University of Gothenburg, 40530 Gothenburg, Sweden; 6Department of Clinical Sciences, Respiratory Medicine and Allergology, Faculty of Medicine, Lund University, 22185 Lund, Sweden; 7Division of Pulmonary, Allergy, and Critical Care Medicine, School of Medicine, University of Pittsburgh, Pittsburgh, PA 15213, USA; 8Multidisciplinary Sleep Unit, Department of Respiratory Medicine, Hospital Universitari de Bellvitge, IDIBELL, University of Barcelona, Hospitalet de Llobregat, 08907 Barcelona, Spain; nsalord@bellvitgehospital.cat; 9Unidade de Hipertensao, Instituto do Coraçao (InCor), Hospital das Clinicas HCFMUSP, Faculdade de Medicina, Universidade de São Paulo, São Paulo 05403-904, Brazil; luciano.drager@incor.usp.br; 10Unidade de Hipertensao, Disciplina de Nefrologia, Hospital das Clinicas HCFMUSP, Faculdade de Medicina, Universidade de São Paulo, São Paulo 05403-900, Brazil; 11Laboratorio de Sono, Divisao de Pneumologia, Instituto do Coracao (InCor), Hospital das Clinicas HCFMUSP, Faculdade de Medicina, Universidade de Sao Paulo, Sao Paulo 05508-220, Brazil; geraldo.lorenzi@gmail.com; 12Sleep Medicine Center, Translational Neuroscience Center, West China Hospital, Sichuan University, Chengdu 610041, China; tangxiangdong@scu.edu.cn

**Keywords:** obstructive sleep apnea, continuous positive airway pressure, lipid profile, total cholesterol

## Abstract

Background: Obstructive sleep apnea (OSA) is associated with dyslipidemia. However, the effects of continuous positive airway pressure (CPAP) treatment on lipid profiles are unclear. Methods: PubMed/Medline, Embase and Cochrane were searched up to July 2021. Randomized controlled trials (RCTs) of CPAP versus controls with ≥4 weeks treatment and reported pre- and post-intervention lipid profiles were included. Weighted mean difference (*WMD*) was used to assess the effect size. Meta-regression was used to explore the potential moderators of post-CPAP treatment changes in lipid profiles. Results: A total of 14 RCTs with 1792 subjects were included. CPAP treatment was associated with a significant decrease in total cholesterol compared to controls (*WMD* = −0.098 mmol/L, 95% CI = −0.169 to −0.027, *p* = 0.007, *I*^2^ = 0.0%). No significant changes in triglyceride, high-density lipoprotein nor low-density lipoprotein were observed after CPAP treatment (all *p* > 0.2). Furthermore, meta-regression models showed that age, gender, body mass index, daytime sleepiness, OSA severity, follow-up study duration, CPAP compliance nor patients with cardiometabolic disease did not moderate the effects of CPAP treatment on lipid profiles (all *p* > 0.05). Conclusions: CPAP treatment decreases total cholesterol at a small magnitude but has no effect on other markers of dyslipidemia in OSA patients. Future studies of CPAP therapy should target combined treatment strategies with lifestyle modifications and/or anti-hyperlipidemic medications in the primary as well as secondary cardiovascular prevention models.

## 1. Introduction

Obstructive sleep apnea (OSA) is one of the most common sleep disorders in adults, affecting nearly 1 billion individuals worldwide [1]. Obesity [2] and dyslipidemia [3] frequently co-exist among patients with OSA. Findings from large cross-sectional studies show increased prevalence of dyslipidemia among OSA patients in a dose-response manner [3]. Several pathological mechanisms including chronic intermittent hypoxia (CIH) [4,5,6,7], sympathetic overactivation [8,9] and sleep fragmentation [10,11] may cause lipid profile dysregulation in OSA. For example, consistent evidence from animal models of OSA has shown that CIH induces fasting dyslipidemia due to activation of the transcription factor sterol regulatory element-binding protein 1 (a master transcription factor that controls lipid metabolism) and overexpression of stearoyl-CoA desaturase-1 (an important downstream enzyme of triglyceride and phospholipid biosynthesis) [12]. Moreover, CIH also impaired clearance of triglyceride-rich lipoproteins, inactivating adipose lipoprotein lipase [13,14].

In theory, continuous positive airway pressure (CPAP), the first-line treatment for OSA, eliminates CIH and therefore is expected to improve dyslipidemia. However, findings of previous studies regarding CPAP effects on each lipid profile have been inconsistent. A meta-analysis including 6 randomized controlled trials (RCTs) with 741 participants showed that CPAP treatment decreased total cholesterol (TC) for 0.15 mmol/L, but no changes in levels of low-density lipoprotein (LDL), high-density lipoprotein (HDL) nor triglyceride (TG) [15] were observed. Another meta-analysis including 6 RCTs with 699 participants showed that CPAP treatment decreased the levels of TC for 0.16 mmol/L, HDL for 0.03 mmol/L and TG for 0.32 mmol/L respectively [16]. However, this meta-analysis included an RCT [17], which was retracted later. Furthermore, these two meta-analyses included crossover studies [17,18,19,20], and both included data from post crossover periods, which is not recommended by the Cochrane Handbook [21], given that this may induce a carry-over effect. Moreover, most RCTs included in these two meta-analyses have relatively short follow-up durations, i.e., over 80% included RCTs with follow-up durations shorter than 24 weeks. Another meta-analysis including 29 cohort studies with 1958 participants showed that CPAP treatment decreased TC and LDL and increased HDL [22]. In addition, findings from the three aforementioned meta-analyses may be influenced by potential publication bias since their article searches were performed using the specific keywords “lipid profile” and might omit some eligible studies with negative findings of CPAP effects on lipid levels. For example, the study by Nguyen et al. in 2010 [23] analyzed lipid profiles as a secondary outcome, but no significant lipid-lowering effect of CPAP was observed. Taken together, the effect of CPAP treatment on lipid profiles in OSA is yet not well understood.

After publication of the three aforementioned meta-analyses, new RCTs with longer follow-up durations have been published. Because both OSA and dyslipidemia are highly associated with cardiometabolic disease [24,25,26], here is a need for an update to address the effect of CPAP on lipid profiles in OSA, which was the rationale for the current study.

## 2. Methods

### Search Strategy and Selection Criteria

This meta-analysis was registered in the Prospective Register of Systematic Reviews (CRD 42020201177) and conducted according to Preferred Reporting Items for Systematic Review and Meta-analysis Protocols [27]. We searched PubMed/Medline, Embase and Cochrane Central Register using the following key terms: “obstructive sleep apnea” AND “continuous positive airway pressure” AND “randomized controlled trial”. The literature search was up to July 2021 with no language restrictions. Appendix A presents the specific search strategies for each database, Figure 1 shows the study selection process and Appendix A lists the included studies.

The review of search results was conducted independently by two researchers (Guo M. and Chen B.). Any inconsistency was adjudicated by the senior author (Li Y.). Studies were eligible if they were: (1) studies recruiting adults with OSA (age ≥ 18 years), (2) RCTs with CPAP and control groups (either sham-CPAP or usual care treatment) with at least 4-week follow-up duration, (3) trials reporting mean and standard deviation (SD)/standard error of any of the 4 plasma lipid profiles (e.g., TC, TG, HDL or LDL) during pre- and post-intervention periods and (4) of any language; there was no restriction. Submitted data from one of the RCTs [28] were provided by the principal investigator of the main study [29]. We excluded studies if they: (1) were designed to examine the effects of anti-hyperlipidemic medications on lipid profiles, (2) involved active weight loss interventions (i.e., vertical banded gastroplasty), (3) included pregnant women, (4) were non-randomized designs or crossover trials or (5) were reviews, editorials, letters or case reports.

## 3. Data Analysis

Two researchers (Guo M. and Chen B.) independently extracted key characteristics (e.g., first author’s name, publication year, sample size, inclusion criteria, percentage of males, percentage of patients on anti-hyperlipidemic medications, et al., Table 1) and target outcomes (e.g., baseline, endpoint and delta values (endpoint minus baseline values) of TC, TG, HDL or LDL) of each trial. Cardiometabolic disease was defined as resistant hypertension, coronary artery disease or diabetes. The Cochrane Collaboration Tool was used to assess quality and risk of bias of included RCTs [30] (Appendix A). In this study, we converted all lipid profiles data from mg/dL to mmol/L for meta-analysis (Appendix A).

## 4. Statistical Analysis

The mean value and SD of change in TC, TG, HDL or LDL for each group were calculated according to the Cochrane Handbook [30]. The weighted mean difference (*WMD*) was used to assess the effect size. The heterogeneity was assessed by *I*^2^. Random or fixed effects model was conducted in the presence (*I*^2^ > 50%) or absence (*I*^2^ ≤ 50%) of heterogeneity, respectively [30]. To explore the potential moderators of change in lipid profiles after CPAP treatment, meta-regression models were performed by using TC, TG, HDL and LDL as the outcomes and using age, male percentage, body mass index (BMI), obesity (the studies in which mean BMI ≥ 30 kg/m^2^), Epworth Sleepiness Scale (ESS), apnea–hypopnea index (AHI), follow-up duration, CPAP compliance and studies that recruited only patients with cardiometabolic diseases patients as independent variables, respectively, according to possible clinical relevance. Due to the limited numbers of included studies, only one independent variable was meta-regressed at each time. Publication bias was assessed by the inspection of funnel plot [31]. The trim and fill method was used to assess the influence of potential publication bias in case of absence of heterogeneity (*I*^2^ ≤ 50%) [21]. Sensitivity analyses were used to test the stability of results. A level of *p*-value <0.05 was considered statistically significant. All statistical analyses were conducted by using Stata (STATA 14.0, Stata Corp, College Station, TX, USA) and R Project (R 3.4.2, R Foundation for Statistical Computing, Vienna, Austria).

## 5. Results

A total of 14 RCTs with 899 patients in the CPAP group and 893 patients in the control group were included. The main characteristics of included studies are presented in Table 1. Among the 1792 subjects included, the mean age was 55.3 ± 7.1 years, baseline BMI was 32.0 ± 5.4 kg/m^2^ and 78.2% were males. The median follow-up duration of included study was 20 weeks (range: 4 to 48 weeks), and median CPAP compliance was 4.8 hours/night (range: 1.9 to 6.0 hours/night).

A total of 11 studies with 1638 patients provided available data of TC levels at pre- and post-CPAP treatment periods. Fixed-effects meta-analysis showed that CPAP treatment resulted in a significant decrease in TC levels (*WMD* = −0.098 mmol/L, 95% CI = −0.169 to −0.027, *p* = 0.007, *I^2^* = 0.0%, Figure 2). No significant publication bias was observed by the inspection of funnel plot (Appendix A). After trim and fill methods, results were similar (Appendix A). Sensitivity analysis confirmed the stability of result that it was not violated after omitting any particular study (Appendix A). Moreover, meta-analyses showed no significant differences in changes in TG (14 studies, 1792 patients, *WMD* = 0.074 mmol/L, 95% CI = −0.056 to 0.205, *p* = 0.264, *I*^2^ = 75.4%), HDL (13 studies, 1572 patients, *WMD* = −0.032 mmol/L, 95% CI = −0.108 to 0.044, *p* = 0.408, *I*^2^ = 92.3%) and LDL (12 studies, 1492 patients, *WMD* = −0.064 mmol/L, 95% CI = −0.185 to 0.056, *p* = 0.296, *I*^2^ = 86.0%, Figure 2) between the CPAP group and control group. No significant publication bias of TG, HDL and LDL was observed by the inspection of funnel plot (Appendix A). Sensitivity analyses confirmed the stability of results for TG, HDL and LDL and that they were not violated after omitting any particular study (Appendix A).

Meta-regression models were conducted to determine the potential moderators of CPAP treatment effect on the changes in lipid profiles. However, no significant association was found between each lipid profile and age, gender, BMI, obesity, ESS, AHI, follow-up duration, CPAP compliance and studies that recruited only patients with cardiometabolic diseases patients (all *p*-value > 0.05, Appendix A). Furthermore, in the sub-group analyses of follow-up duration with 12, 16 and 20 weeks as cut-off points, respectively, no significant between-group difference of any lipid profile was observed (all *p*-value > 0.05). Moreover, sub-group analyses by different diagnostic criteria for OSA (using AHI versus using oxygen desaturation index) showed no significant differences between the two groups in each lipid profile (all *p*-value > 0.1).

## 6. Discussion

This is an updated meta-analysis including 14 RCTs with a total of 1792 participants (as many as twice the number of RCTs and participants compared to the previous ones). Our findings indicate that CPAP treatment decreases TC in adults with OSA, though only by at a relatively small magnitude. However, there is no effect of CPAP treatment on TG, HDL and LDL levels in adults with OSA.

In 2014, two meta-analyses including RCTs examined the effect of CPAP treatment on lipid profiles in OSA and found that CPAP treatment promoted decreased TC for 0.15 and 0.16 mmol/L, respectively [15,16]. In the current study, we have a similar finding of decreased TC after CPAP treatment, but this effect was of a smaller magnitude. The lower rate of decrease of TC in the current study may be partially interpreted by the fact that we used a less restricted search strategy and included more studies reporting negative findings of CPAP effects on changes in lipid profiles. As for other lipid profiles (i.e., TG, HDL and LDL), no changes after CPAP treatment were observed, which is consistent with the previous meta-analyses [15,16,22], suggesting CPAP therapy promotes limited improvement of dyslipidemia in OSA.

The underlying mechanisms for reduction of TC levels after CPAP treatment in OSA are unclear. First, it could be associated with the improvement of CIH by CPAP. CIH, one of the main pathological conditions in OSA, upregulates the pathways of hepatic liver biosynthesis in a fasting state [4] and delays post-prandial lipid clearance [5,13,14] through inducing activation of the enzyme of triglyceride and phospholipid biosynthesis [12], excessive production of reactive oxygen species [6] and low-grade inflammation [7], which has been proposed as one of the main mechanisms for OSA-inducing hyperlipidemia. Since TC is one of the first components to respond to the reduction in oxidative stress associated with OSA treatment [32], it is expected that TC levels decrease after CPAP treatment. Second, decreased levels of sympathetic activity [33], cortisol [34] and insulin [35] after CPAP treatment may result in decreased lipid levels. Increased levels of norepinephrine and cortisol, as well as insulin resistance, have been noted to collectively stimulate lipolysis in adipose tissue and induce syntheses of hepatic fatty acid and lipid profiles [8]. Third, it could be associated with the improvement of sleep continuity by CPAP treatment because dyslipidemia may result from sleep fragmentation [11] caused by apneic events. Finally, it could be associated with improvement of fatigue and excessive daytime sleepiness after CPAP treatment [36,37], which may result in increased levels of physical activity. However, a 10-year follow-up cohort study among elderly suggested that worsening of nocturnal oxygen desaturation was independent of changes in circulating lipids and not influenced by lipid-lowering treatments [38]. However, the changes in blood pressure remained associated with waist/hip and LDL/HDL ratios. Taken together, besides sleep apnea, other factors such as age, blood pressure and central obesity may affect lipid levels. However, our findings of meta-regression show that age, cardiometabolic diseases and obesity do not moderate the effect of CPAP on lipid levels. Future studies should be conducted to examine the underlying mechanisms for limited effects of CPAP on lipid levels in patients with OSA.

In the previous two meta-analyses with RCTs examining CPAP effects on lipid profiles, the findings of moderators for CPAP effects on lipid profiles are inconsistent. For example, one reported that a better lipid-lowering effect was observed in studies with longer follow-up duration [16], while the other reported an opposite finding [15]. In the current study, meta-regression models show that post-CPAP treatment changes in TC, as well as other lipid profiles, are not moderated by age, sex, BMI, daytime sleepiness, the severity of OSA, follow-up duration or CPAP compliance, suggesting no single moderator influences the main outcome for lipid profiles.

Our study has several clinical implications. The findings of a decrease in TC suggest that CPAP treatment improves lipid metabolism in OSA. However, such relatively small decrement of TC (−0.098 mmol/L; 3.793 mg/dL) could be the result of slightly decreased LDL and HDL after CPAP treatment. Its clinical significance should be interpreted cautiously. Future longitudinal studies should examine the clinical implications regarding decreasing cardiovascular risk at the relatively small magnitude decrease in TC. Moreover, our findings show no effects of CPAP treatment on HDL and LDL. Thus, it appears that CPAP treatment alone does not improve the lipid profiles in OSA patients with dyslipidemia, and CPAP should be combined with lifestyle modifications and anti-hyperlipidemic medications [39]. Of note, one of the RCTs addressed the effect of CPAP in patients with OSA and coronary artery disease who were already on anti-hyperlipidemic medication without any additional improvement [28,29]. Notwithstanding, the combined effect of CPAP and anti-hyperlipidemic medication might be more effective among patients with OSA free from cardiometabolic disease at baseline compared to the effects in patients who already have developed a cardiometabolic disease. In addition, barbed repositioning pharyngoplasty has been shown to improve chronic inflammation and cardiometabolic disease, which may be regarded as one efficient intervention for obese OSA patients [40,41].

The current study has some strengths to be addressed. Comparing to the previous two meta-analyses with RCTs [15,16], we included as many as twice the number of RCTs, of which seven have relatively long follow-up durations (24-to-48 weeks). Some limitations need to be acknowledged. Lipid levels are associated with diet, medications (i.e., anti-hyperlipidemic medications, insulin and beta-blockers), daily physical activity and lifestyle [42]. Unfortunately, such confounders might not be well-controlled in the current study since most of the included RCTs were not specifically designed to evaluate lipid profiles and did not provide information regarding these confounders. Furthermore, some participants using anti-hyperlipidemic medications were included, and only five studies reported the percentage of using anti-hyperlipidemic medications, which does not allow us to eliminate its confounding effect. Future studies should fully consider the aforementioned confounding effects when examining the effects of CPAP on lipid profile. Moreover, CPAP compliance in this meta-analysis was based on the mean compliance for each study but not for each patient. Therefore, the non-significant relationship between changes in lipid profiles and CPAP compliance in meta-regression models should be interpreted cautiously and be examined in future studies. Finally, since a great part of the included participants were males (78.2%), sex-stratified designed studies are also needed.

## 7. Conclusions

CPAP treatment decreases TC at a small magnitude in adults with OSA. Since TC is a strong predictor for cardiometabolic diseases, our findings indicate that CPAP combined with lipid-lowering drugs are warranted for OSA patients with dyslipidemia. Future studies should be conducted to explore the potential mechanisms for CPAP treatment effects on lipid profiles.

## Figures and Tables

**Figure 1 jcm-11-00596-f001:**
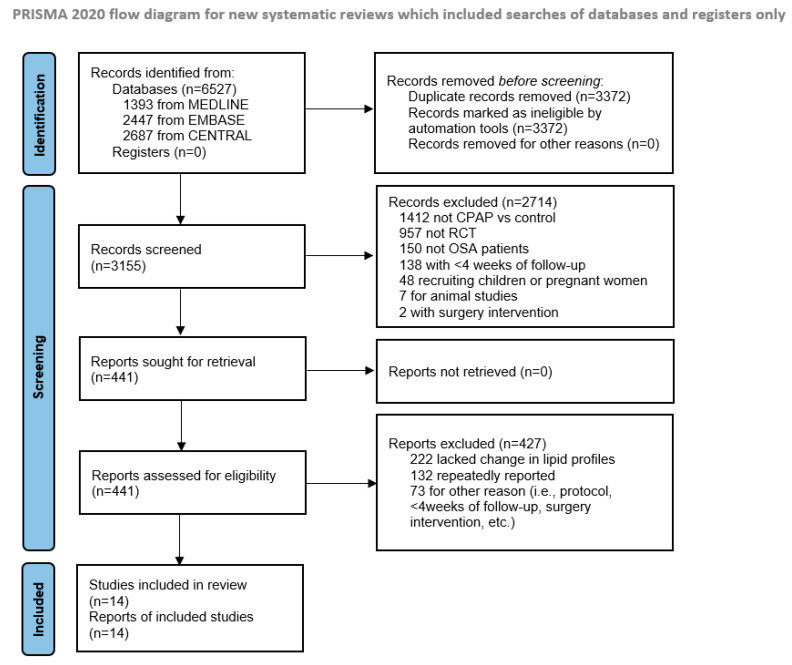
Flow chart of literature search. CPAP = continuous positive airway pressure, OSA = obstructive sleep apnea, RCT = randomized controlled trial.

**Figure 2 jcm-11-00596-f002:**
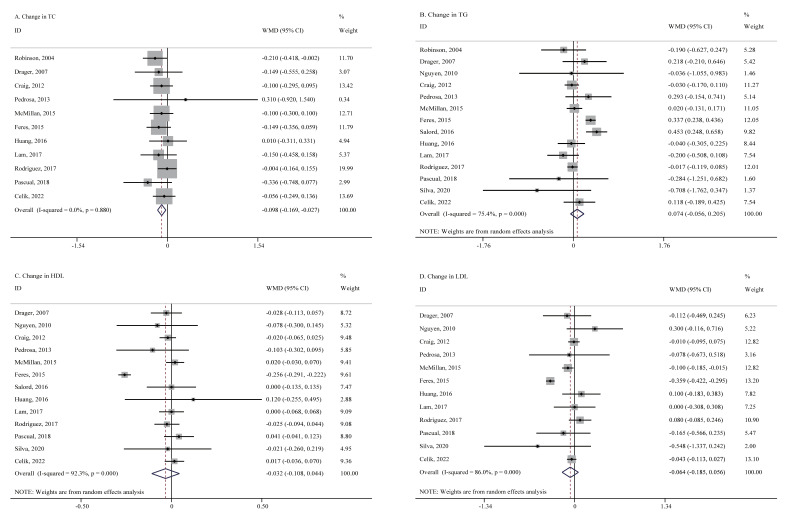
Forest plots for changes in lipid profiles after CPAP treatment. CPAP = continuous positive airway pressure, HDL = high-density lipoprotein, LDL = low-density lipoprotein, TC = total cholesterol, TG = triglyceride, WMD = weighted mean difference. Panel (**A**) Change in TC. Panel (**B**) Change in TG. Panel (**C**) Change in HDL. Panel (**D**) Change in LDL.

**Table 1 jcm-11-00596-t001:** Characteristics of the 14 included studies.

Study (First Author, Year)	N, CPAP	N, Control	Inclusion Criteria	Cardiometabolic Disease	Country	Sham-CPAP Controlled	Age (Year)	Male (%)	BMI (kg/m^2^)	Follow-Up (Week)	Baseline ESS	AHI (Event/Hour)	CPAP Compliance (Hour/Night)	Use of Anti-hyperlipidemic Medications (%)
Robinson, 2004	108	112	ESS > 9; ODI > 10; Male	No	UK	Yes	45.5	100	29.9	4	16.2	38.7	5	-
Drager, 2007	12	12	AHI > 30; BMI ≤ 35; Age < 60; Non-HT; Non-diabetes; Male	No	Brazil	No	53.4	100	29.8	16	13.5	59	6	-
Nguyen, 2010	10	10	AHI ≥ 15; ESS > 10	No	USA	Yes	57.7	90	32.4	12	-	35.2	5.1	-
Craig, 2012	172	174	ODI > 7.5	No	UK, Canada	No	56	78	32	24	8	13.4	2.65	-
Pedrosa, 2013	19	16	AHI ≥ 15; RHT	Yes	Brazil	No	69.9	77	33.8	24	10	29	6.01	-
McMillan, 2015	114	117	Age ≥ 65; ODI > 7.5; ESS ≥ 9	No	UK	No	53.7	81.7	30.2	48	11.6	28.7	1.9	-
Feres, 2015	22	23	AHI > 5; BMI ≤ 40	No	Brazil	Yes	46.6	-	47.4	24	-	40.2	-	-
Salord, 2016	42	38	AHI > 30; Non-diabetes; BMI ≥ 35 with obesity co-morbidity or BMI ≥ 40	No	Spain	No	62.1	27.5	23.0	12	7.9	60.8	5.4	6.3
Huang, 2016	37	33	AHI ≥ 15; Newly diagnosed coronary artery disease; Non-diabetes; BMI < 25; ESS < 14	Yes	China	No	54.5	83.3	30.8	48	9	28.9	4.2	100
Lam, 2017	32	32	AHI ≥ 15; Diabetes	Yes	Hongkong China	No	57.1	81	33.7	12	7.5	45.3	2.5	-
Rodriguez, 2017	151	156	AHI ≥ 15; Female	No	Spain	No	54.8	0	33	12	9.8	32	4.8	34.5
Pascual, 2018	30	27	AHI > 20; Erectile dysfunction	No	Spain	No	49.4	100	35.8	12	10.2	51.6	5.3	25.3
Silva, 2020	31	23	AHI: 5~15; Age < 65; BMI < 35	No	Brazil	No	47.4	51.9	28.4	48	-	9.7	3.8	-
Celik, 2022	94	102	AHI ≥ 15; ESS < 10; Coronary artery disease	Yes	Sweden	No	66	84.2	28.3	38	5.5	28.8	3.3	95

AHI = apnea–hypopnea index; BMI = body mass index; CPAP = continuous positive airway pressure; DBP = diastolic blood pressure; ESS = Epworth Sleepiness Scale; HT = hypertension; ODI = oxygen desaturation index; SBP = systolic blood pressure; RCT = randomized controlled trial; RHT = resistant hypertension. -indicates that the value was not reported.

## Data Availability

Data that underlie the results reported in this article can be obtained by contacting the corresponding author; s_liyun@stu.edu.cn.

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
