# Peer review of "Effect of Continuous Positive Airway Pressure on Lipid Profiles in Obstructive Sleep Apnea: A Meta-Analysis"

_jcm, 2022, doi:10.3390/jcm11030596_

Round 1

Reviewer 1 Report

In this meta-analysis, Chen and coworkers assess the effect of CPAP on lipid profile in OSA patients. The study is an update of older metanalyses and is worth a deeper look.

My concerns are listed as follows:

  1. The effect of CPAP on total cholesterol is practically negligible. I understand that it hit the statistical threshold, however, clinically, a cholesterol change of 0.1 mmoL/L (less than 2 mg/dL!) would be disregarded by anyone. In addition, the significant decrement of total cholesterol could be the result of slightly decreased LDL and HDL, which would detract even more from the importance of the finding. The author should discuss this more where appropriate, maybe tuning down the emphasis on the results in the discussion (adding the comparative measure in mg/dL to help the reader realize the magnitude of the cholesterol decrease), especially in the cardiovascular risk paragraph.
  2. At first, I thought it was odd that CPAP compliance did not moderate the effect on cholesterol. In light of the small cholesterol reduction, I can re-considerate my opinion, however, the authors should consider discussing potential bias in how the CPAP compliance was measured; in fact, if they sell the message that CPAP does not have a dose dependent effect on the outcome, all the pathophysiological issues that CPAP addresses (and which the authors describe) would no longer be valid.
  3. I did not grasp the matter of the follow up duration. It seems an issue for the authors that previous studies had a short follow up time (<24 weeks, even though in the results of this study, only 16 weeks is mentioned). However, to my understanding, this trial should measure the effect of CPAP from baseline (time x) to end of treatment (time y). The follow up time of any duration (time z) could be of interest, but partially out of the scope of the study. I am concerned for two reasons here: a) the authors did not describe the treatment time and 2) the authors say that, when the post treatment outcomes were reported more than one time, the “longest” (i.e. furthest in time?) value was recorded (thus, closer to time z than to time y). This could hide a bias. I suggest reconsidering the values to be taken the closest possible to end of treatment time and/or setting up a time domain analysis (effect of cholesterol vs time x, y, z’s).
  4. Line 105: the authors DID NOT specify their search criteria for cholesterol/lipid profile.
  5. Maybe reference 29 and 30 should be the other way around? I also personally have never seen a review that quotes unpublished data, since they have not gone through peer review yet. I wonder what the results of the metanalysis would be without accounting for the unpublished study.
  6. It is odd that only 5 studies recorded the usage of anti-hyperlipemics. Did any of them assess diet (Kcal/meal) or physical exercise? In this regard, line 270 is not entirely clear.
  7. The authors mention in the figure and in the results that they ultimately selected 14 studies, but then, in the results and in another figure (3.4, which is also too small, I suggest reformatting), they only talk about 11. What happened to the other 3?

Additionally, I have other minor suggestions:

  1. Lines 52-59 are part of some preconstructed template and need to go.
  2. Line 66: lipid profile.
  3. Line 68: the authors better say what this protein does since few would know that.
  4. Line 92: outcome.
  5. In the introduction, the authors say that there is inconsistency on the effect of CPAP on lipid profile, however it seems that there is a unidirectional signal from all the studies in favor of CPAP.
  6. Line 92: references are missing.
  7. It is not clear what the other reason for study exclusion were: what does “protocol” or “surgery mean”? Were 4-week follow up studies not already excluded in the previous step?
  8. Were potential moderators selected according to possible clinical relatedness?
  9. Lines 154-156 are redundant with table.
  10. Table 2 is a little invasive. Since it only illustrates negative findings, I recommend moving it to the supplement.
  11. Lines 213-215 are a repetition of what said in the introduction.

Author Response

Reviewer 1

  1. The effect of CPAP on total cholesterol is practically negligible. I understand that it hit the statistical threshold, however, clinically, a cholesterol change of 0.1 mmoL/L (less than 2 mg/dL!) would be disregarded by anyone. In addition, the significant decrement of total cholesterol could be the result of slightly decreased LDL and HDL, which would detract even more from the importance of the finding. The author should discuss this more where appropriate, maybe tuning down the emphasis on the results in the discussion (adding the comparative measure in mg/dL to help the reader realize the magnitude of the cholesterol decrease), especially in the cardiovascular risk paragraph.

Response: We agree with the reviewer that decrease in total cholesterol may be the result of slightly decreased LDL and HDL and its clinical significance need to be discussed cautiously. We have now modified the discussion regarding clinical implications in the revised manuscript: However, such relatively small decrement of TC (-0.098 mmol/l; 3.793 mg/dl) could be the result of slightly decreased LDL and HDL after CPAP treatment. Its clinical significance should be interpreted cautiously. Future longitudinal studies should examine the clinical implications regarding decreasing cardiovascular risk at the relatively small magnitude decrease in TC.

  1. At first, I thought it was odd that CPAP compliance did not moderate the effect on cholesterol. In light of the small cholesterol reduction, I can re-considerate my opinion, however, the authors should consider discussing potential bias in how the CPAP compliance was measured; in fact, if they sell the message that CPAP does not have a dose dependent effect on the outcome, all the pathophysiological issues that CPAP addresses (and which the authors describe) would no longer be valid.

 Response: CPAP compliance in this meta-analysis was based on the mean compliance for each study, but not for each patient. Thus, the non-significant relationship between changes in lipid profiles and CPAP compliance in meta-regression models should be interpreted cautiously and be examined in future studies. We have now added the aforementioned discussion in Disscussion and list as a limitation in the revised manuscript.

  1. I did not grasp the matter of the follow up duration. It seems an issue for the authors that previous studies had a short follow up time (<24 weeks, even though in the results of this study, only 16 weeks is mentioned). However, to my understanding, this trial should measure the effect of CPAP from baseline (time x) to end of treatment (time y). The follow up time of any duration (time z) could be of interest, but partially out of the scope of the study. I am concerned for two reasons here: a) the authors did not describe the treatment time and 2) the authors say that, when the post treatment outcomes were reported more than one time, the “longest” (i.e. furthest in time?) value was recorded (thus, closer to time z than to time y). This could hide a bias. I suggest reconsidering the values to be taken the closest possible to end of treatment time and/or setting up a time domain analysis (effect of cholesterol vs time x, y, z’s).

Response: We appreciate this comment and agree with the reviewer’s concerns. In the previous version of manuscript, it was stated that “When the post-intervention values of lipid profiles were reported for more than one-time point, the longest duration were used”. In fact, the post-intervention lipid profiles were reported only at one-time point in all the included studies. In order to be more clear, we therefore have now deleted this sentence.

  1. Line 105: the authors DID NOT specify their search criteria for cholesterol/lipid profile.

Response: In this study, we did not use specific searching keyword of lipid profile, which was aimed to reduce publishing bias. The reason has been presented in 2nd paragraph of Methods: we used a less restricted search strategy and included more studies reporting negative findings of CPAP effects on changes in lipid profiles. For example, the previous two meta-analyses included “lipid profiles” as a key term for literature searching, which may have resulted in missing some studies that addressed the effect of CPAP treatment on lipid profiles as a secondary outcome (i.e., Patricia K Nguyen et al., Nasal continuous positive airway pressure improves myocardial perfusion reserve and endothelial-dependent vasodilation in patients with obstructive sleep apnea, Journal of Cardiovascular Magnetic Resonance, 2010).

  1. Maybe reference 29 and 30 should be the other way around? I also personally have never seen a review that quotes unpublished data, since they have not gone through peer review yet. I wonder what the results of the metanalysis would be without accounting for the unpublished study.

Response: Both references are from a same published trial (Peker Y et al., Effect of Positive Airway Pressure on Cardiovascular Outcomes in Coronary Artery Disease Patients with Nonsleepy Obstructive Sleep Apnea. The RICCADSA Randomized Controlled Trial, Am J Respir Crit Care Med, 2016). It is a sub-study of RICCADSA trial for the unpublished reference. The principal investigator of the unpublished reference is co-author of the current meta-analysis. And currently the cited reference has gone through peer-review and been accepted for publication in the Journal of Clinical Medicine. We will update the reference accordingly when the official DOI-number is given soon.

  1. It is odd that only 5 studies recorded the usage of anti-hyperlipemics. Did any of them assess diet (Kcal/meal) or physical exercise? In this regard, line 270 is not entirely clear.

Response: In this study, most of the included studies were not specifically designed to determine the effect of CPAP on lipid profile. Thus, it is conceivable that only a few studies provide the usage of anti-hyperlipemics. Furthermore, no study provided data regarding diet or exercise. In this regard, we have now rephrased the limitation section as below: “Lipid levels are associated with diet, medications (i.e., antihyperlipidemic medications, insulin and beta-blockers), daily physical activity and lifestyle. Unfortunately, such confounders might not be well controlled in the current study since most of the included RCTs are not specifically designed to evaluate lipid profiles and did not provide information regarding these confounders. Future studies should fully consider the confounding effects of medications, daily physical activity and lifestyle when examining the effects of CPAP on lipid profile”.

  1. The authors mention in the figure and in the results that they ultimately selected 14 studies, but then, in the results and in another figure (3.4, which is also too small, I suggest reformatting), they only talk about 11. What happened to the other 3?

Response: We have now reformatted and uploaded the Figures as a separate PDF document to make it more clear. Regarding the number of studies, we included overall 14 RCTs in this meta-analysis. However among them, 11 reported total cholesterol, 14 reported triglyceride, 13 reported HDL, and 12 reported LDL. These exact numbers has been presented in the 2nd paragraph of Result section.

Additionally, I have other minor suggestions:

  1. Lines 52-59 are part of some preconstructed template and need to go.

 Response: We have now deleted this part.

  1. Line 66: lipid profile.

 Response: It has been corrected.

  1. Line 68: the authors better say what this protein does since few would know that.

 Response: The transcription factor sterol regulatory element-binding protein 1 is a master transcription factor that controls lipid metabolism. Its activation is linked to dyslipidemia. We have now rephrased this sentence as “For example, consistent evidence from animal models of OSA has shown that CIH induces fasting dyslipidemia due to activation of the transcription factor sterol regulatory element-binding protein 1 (a master transcription factor that controls lipid metabolism) and overexpression of stearoyl-CoA desaturase-1 (an important downstream enzyme of triglyceride and phospholipid biosynthesis)”.

  1. Line 92: outcome.

Response: It has been corrected.

  1. In the introduction, the authors say that there is inconsistency on the effect of CPAP on lipid profile, however it seems that there is a unidirectional signal from all the studies in favor of CPAP.

Response: We agree with the reviewer that all 3 previous meta-analyses supported the improving effect of CPAP on dyslipidemia. However, there is inconsistency for CPAP effect on each lipid profile. For instance, significant change in HDL was only found in 2 studies and change in LDL was found in one study. We have now revised the corresponding sentence as “However, findings of previous studies regarding CPAP effects on each lipid profile have been inconsistent.”

  1. Line 92: references are missing.

 Response: The reference (Patricia K Nguyen et al., Nasal continuous positive airway pressure improves myocardial perfusion reserve and endothelial-dependent vasodilation in patients with obstructive sleep apnea, Journal of Cardiovascular Magnetic Resonance, 2010) has been cited as Number 24 in the revised manuscript.

  1. It is not clear what the other reason for study exclusion were: what does “protocol” or “surgery mean”? Were 4-week follow up studies not already excluded in the previous step?

Response: In this meta-analysis, the eligible criteria included studies with at least 4 week intervention and studies without active weight loss interventions like vertical banded gastroplasty. However, because some studies did not reported follow-duration in the abstract, we cannot identify all eligible studies through screening of titles and abstracts. For instance, the study (Henke KG et al., Effect of nasal continuous positive airway pressure on neuropsychological function in sleep apnea-hypopnea syndrome. A randomized, placebo-controlled trial, Am J Respir Crit Care Med, 2001) did not report intervention duration in title and abstract, but provided their competed study protocol in the full-text, which allowed us to confidently exclude it.

  1. Were potential moderators selected according to possible clinical relatedness?

Response: If I understood correctly, the reviewer is asking why we selected those variables for meta-regression. In meta-regression models, we selected the common potential moderators of lipid levels according to possible clinical relevance, including age, gender, BMI, obesity, daytime sleepines (ESS score), sleep apnea severity (AHI), follow-up duration, CPAP compliance, cardiometabolic disease. We have now included this reason for selecting variables for meta-regression in the section of Methods of the revised manuscript.

  1. Lines 154-156 are redundant with table.

Response: We have now deleted the redundant context in the revised manuscript.

  1. Table 2 is a little invasive. Since it only illustrates negative findings, I recommend moving it to the supplement.

Response: We appreciated this suggestion and have now moved the Table 2 into supplemental Table (Table S6)

  1. Lines 213-215 are a repetition of what said in the introduction.

 Response: We have now deleted the repeated context at original line 213-215.

Reviewer 2 Report

the first sentences of the introduction are the informations. please

remove

  • despite obstructive sleep apnea present as primary treatment CPAP, barbed surgery in subjects with collapse of the lateral walls has shown excellent results on cardiometabolic risk, reducing the chronic inflammatory state. please cite doi:10.1016/j.amjoto.2021.103197 and doi:10.1007/s00405-016-4290-0
  • an interesting paper reported among the 146 participants for whom there were follow-up data, those assigned to weight loss only and those assigned to the combined interventions had reductions in CRP levels, insulin resistance, and serum triglyceride levels. None of these changes were observed in the group receiving CPAP alone. Blood pressure was reduced in all three groups. please discuss and cite doi:10.1056/NEJMoa1306187
  • line 104, why not scielo among database?
  • please use the latest prisma flow diagram
  • trim and fill method was performed?
  • A study suggested that the observed worsening of nocturnal oxygen desaturation after 10 years in the elderly was independent of the change in circulating lipids and not affected by lipid-lowering treatments. However, the change in blood pressure remained associated with aging, waist / hip ratio, and LDL-C / HDL-C ratio. However aging is strictly related to osa severity in elderly patients. please discuss and cite doi:10.1016/j.sleep.2017.07.028 and doi:10.1056/NEJMoa1306187

Author Response

Reviewer 2

  1. The first sentences of the introduction are the informations. Please remove.

Response: We appreciate the reviewer’s suggestion. Now these sentences have been removed in revised manuscript.

  1. Despite obstructive sleep apnea present as primary treatment CPAP, barbed surgery in subjects with collapse of the lateral walls has shown excellent results on cardiometabolic risk, reducing the chronic inflammatory state. please cite doi:10.1016/j.amjoto.2021.103197 and doi:10.1007/s00405-016-4290-0

Response: We appreciated the reviewer’s suggestion. We have now included the relevant contents in discussion and cited these references in the revised manuscript: “In addition, barbed repositioning pharyngoplasty has been shown improving effect on chronic inflammation and cardiometabolic disease, which may be also regarded as one efficient intervention for OSA”.

  1. An interesting paper reported among the 146 participants for whom there were follow-up data, those assigned to weight loss only and those assigned to the combined interventions had reductions in CRP levels, insulin resistance, and serum triglyceride levels. None of these changes were observed in the group receiving CPAP alone. Blood pressure was reduced in all three groups. please discuss and cite doi:10.1056/NEJMoa1306187

Response: We appreciated the reviewer’s suggestion. We have now included relevant discussion and cited this reference in the revised manuscript: “It appears that CPAP treatment alone does not improve the lipid profiles in OSA patients with dyslipidemia, and CPAP should be combined with life-style modifications and antihyperlipidemic medications”.

  1. Line 104, why not scielo among database?

 Response: In this meta-analysis, we did not search the SciELO database. According to the section of bibliographic databases in Cochrane Handbook, the three bibliographic databases generally considered to be the most important sources to search for reports of trials – CENTRAL, MEDLINE and EMBASE. We therefore only searched these three databases.

  1. Please use the latest prisma flow diagram

 Response: We appreciate this suggestion and have now used the latest flow diagram.

  1. Trim and fill method was performed?

 Response: We did not perform trim and fill method. We have now included results of trim and fill method. Because the trim and fill method is known to perform poorly in the presence of substantial between-study heterogeneity according to the section of trim and fill in Cochrane Handbook, we only use this method in case of absence of heterogeneity (I2≤50%). The results remain similar and significant after trim and fill. Please see Figure S2 and Table S4.

  1. A study suggested that the observed worsening of nocturnal oxygen desaturation after 10 years in the elderly was independent of the change in circulating lipids and not affected by lipid-lowering treatments. However, the change in blood pressure remained associated with aging, waist / hip ratio, and LDL-C / HDL-C ratio. However aging is strictly related to osa severity in elderly patients. please discuss and cite doi:10.1016/j.sleep.2017.07.028 and doi:10.1056/NEJMoa1306187

 Response: We appreciated this suggestion. We have now added relevant discussion and cited these two references in Discussion section of the revised manuscript: “However, a 10-year follow-up cohort study among elderly suggested that worsening of nocturnal oxygen desaturation was independent of change in circulating lipids, and not influenced by lipid-lowering treatments. However, the changes in blood pressure remained associated with waist/hip and LDL/HDL ratios. Taken together, besides sleep apnea, other factors, ie, age, blood pressure and central obesity may affect lipid levels. However, findings of meta-regression shows that age, cardiometabolic diseases and obesity do not moderate the effect of CPAP on lipid levels. Future studies should be conducted to examine the underlying mechanisms for limited effects of CPAP on lipid levels in patients with OSA.”

Reviewer 3 Report

In this study, 14 RCTs were used for meta-analysis. The objective of this study is to understand whether lipid profile would be changed among OSA patients treatment from CPAP used. Although there have been similar studies in the past, the results are still inconsistent. There are still a few questions I would like to ask the authors:

  1. Seven of the 14 studies had a follow-up time greater than 24 weeks. Is this result very different from the comparison between the 15th and 16th references (Xu et al., 2014; Lin et al., 2015)? Is it possible to use a stratified analysis method to analyze different follow-up times and to clarify the effect of follow-up time?
  2. The study of Craig and Mcmillan (Craig et al., 2012; Mcmillan et al., 2015) diagnosed OSA patients as ODI>7.5, and Robinson's study (Robinson et al., 2004) also diagnosed OSA patients as ODI >10, which is different from the criterion of using AHI for judgment the OSA. Is this criteria of OSA diagnosis will affect the analysis results?
  3. Please include all research included in the review of meta-analysis process in the reference list.
